

# Geometric remapping of particle distributions in the Discrete Element Model for Sea Ice (DEMSI v0.0)

Adrian K. Turner[1], Kara J. Peterson[2], and Dan Bolintineanu[2]

[1]Theoretical Division, Los Alamos National Laboratory, Los Alamos, NM, USA
[2]Sandia National Laboratories, Albuquerque, NM, USA

**Correspondence:** Adrian K. Turner (akt@lanl.gov)

**Abstract.** A new sea ice dynamical core, the Discrete Element Model for Sea Ice (DEMSI), is under development for use in coupled Earth system models. DEMSI is based on the discrete element method, which models collections of ice floes as interacting Lagrangian particles. In basin-scale sea ice simulations the Lagrangian motion results in significant convergence and ridging, which requires periodic remapping of sea ice variables from a deformed particle configuration back to an undeformed initial distribution. At the resolution required for Earth system models we cannot resolve individual sea ice floes, so we adopt the sub-gridscale thickness distribution used in continuum sea ice models. This choice leads to a series of hierarchical tracers depending on ice fractional area or concentration that must be remapped consistently. The circular discrete elements employed in DEMSI help improve the computational efficiency at the cost of increased complexity in the effective element area definitions for sea ice cover that are required for the accurate enforcement of conservation. An additional challenge is the accurate remapping of element values along the ice edge, the location of which varies due to the Lagrangian motion of the particles. In this paper we describe a particle-to-particle remapping approach based on well-established geometric remapping ideas that enforces conservation, bounds-preservation, and compatibility between associated tracer quantities, while also robustly managing remapping at the ice edge. One element of the remapping algorithm is a novel optimization-based flux correction that enforces concentration bounds in the case of non-uniform motion. We demonstrate the accuracy and utility of the algorithm in a series of numerical test cases.

## 1 Introduction

Sea ice, the frozen surface of the ocean at high latitudes, forms an important component of the Earth climate system. Sea ice moderates the exchange of heat, mass, and momentum between the ocean and the atmosphere. The high albedo of sea ice has a significant effect on planetary reflectivity, and can help drive the polar amplification of climate change through an albedo feedback mechanism (Ingram et al., 1989), while the rejection of salt during sea-ice formation helps drive the thermohaline circulation (Killworth, 1983). The sea ice components of current global climate models use a continuum Eulerian formulation, and use either structured grids (e.g. CICE: Hunke et al. (2015), or LIM3: Rousset et al. (2015)) or unstructured meshes (e.g. MPAS-Seaice: Petersen et al. (2019)). Continuum Lagrangian sea-ice models have also been developed, such as neXtSIM (Rampal et al., 2016) which uses a moving triangulation as its mesh. The Discrete Element Method (DEM) is an alternate





Lagrangian method where the motion and collisions of finite sized particles are simulated. Several DEM sea ice models have been developed for process scale simulations, such as for sea ice ridging and deformation (Hopkins, 1994), the interaction between sea ice and solid structures (e.g. Tuhkuri and Polojärvi, 2018), the interaction between sea ice and waves (e.g. Xu et al., 2012; Herman, 2017), floe clustering (Herman, 2011), and channel flow (e.g. Gutfraind and Savage, 1998). Basin scale DEM sea ice models have been developed as well (Hopkins, 2004), and have been used to study the formation of the floe size

distribution through fracture processes between elements (Hopkins and Thorndike, 2006).

Ice convergence and ridging present a unique set of issues for sea-ice DEM models that are not present in traditional DEM applications. During ridging ice area is reduced as ice thickness increases. Capturing the relevant physics of this process using DEM sea ice models has proven to be a challenge. In the work of Hopkins (2004), the authors represented the ridging process through a remapping scheme of overlapping converging neighboring elements. In this model, every 24 hours the spatial

distribution of elements is remapped back to the initial distribution of elements. Elements that overlap thicken after remapping representing the ridging process. Without this remapping, element overlap would increase indefinitely, producing a highly interpenetrated element distribution for which determination of element contacts would be difficult, as would interpretation of the sea ice state at any particular point. Alternatively, instead of using overlapped elements to represent ridging, elements could be made to shrink instead. Elements with arbitrarily small radii, however, introduce computational challenges with regard

to efficient contact searching, time step size, and contact model formulation (Hopkins, 2004; Shire et al., 2020). Both these considerations, therefore, require long duration DEM simulations of sea ice deformation to periodically perform a remapping of the model elements to an undeformed distribution.

Computational performance is another important consideration for long duration DEM sea ice simulations. One of the most computationally expensive parts of a DEM model is the detection of collisions between neighboring elements. Hopkins (2004)

use polygonal elements and so require a computationally expensive algorithm (Preparata and Shamos, 1985) to detect the collision between elements and to calculate the collision point. Collision detection between circular elements, on the other hand, is much less computationally expensive since for circular elements only a comparison between the element separation and the element radii is needed to determine if a collision has occurred. In this regard, in the work which follows, we explore how to represent sea ice in a Hopkins (2004)-like DEM model using more computationally efficient circular elements. Within

this framework, we also investigate how to perform proper remapping of elements, necessitated by the ridging process and required for the successful development of a global DEM sea ice model performing long duration simulations.

Considerable research has gone into developing conservative and bounds-preserving remapping methods for transferring scalar quantities between two grids. In the geometric approach, a reconstruction of the conserved quantity is integrated over intersections of overlapping cells between the source and target mesh. In the climate modeling community this class of al-

gorithm has been specialized for remapping between structured and unstructured spherical grids (Jones, 1999; Ullrich et al., 2009). Similar remapping algorithms have been developed for use in Arbitrary Lagrangian-Eulerian methods (Margolin and Shashkov, 2003) as well as the remapping step of semi-Lagrangian or incremental remapping schemes for transport (Dukowicz and Baumgardner, 2000; Lipscomb and Hunke, 2001; Lipscomb and Ringler, 2005; Lauritzen et al., 2010). For second-order and higher remapping schemes, a form of limiting must be done to preserve physical bounds on the remapped field. Bounds





preservation can be achieved through limiting the gradient of the reconstruction (Dukowicz and Baumgardner, 2000; van Leer, 1979), applying a flux-correction algorithm (Liska et al., 2010), or applying an optimization-based correction (Bochev et al., 2013, 2014).

In this paper we present a method that builds on standard geometric remapping approaches while addressing the unique challenges associated with remapping for a DEM sea ice model using circular elements. These challenges include defining

consistent areas for enforcing conservation, addressing monotonicity errors due to element overlap under non-uniform motion, and enabling accurate reconstructions at the ice edge. In Section 2 we describe the representation of sea ice in our model and introduce an effective element area needed to represent sea ice with 100% ice concentration. In Section 3 we describe our remapping algorithm including a novel optimization-based flux correction for the case of non-uniform motion and methods for ensuring accurate remap at the ice edge. In Section 4 we present numerical examples that demonstrate the method is robust

and achieves second-order accuracy, tracer compatibility, conservation, and bounds preservation.

## 2   Representation of sea ice with circular elements

While using circular elements in DEM models is computationally efficient, they present a unique challenge for the representation of sea ice. Unlike polygonal elements that can be made to completely tessellate a region (e.g. with a Voronoi tessellation), circular elements, in general, cannot completely cover a region, preventing them from directly representing sea ice cover with

100% concentration. In order to represent 100% ice cover with circular elements we associate an effective area, $e$, with each element. This area, potentially larger than the geometric area of the circular element, represents the area of sea ice and open water associated with that element. We determine this initial effective element area from a radical Voronoi tessellation (Imai et al., 1985) of the elements. Also called a power, or Laguerre–Voronoi diagram, this is a Voronoi tessellation weighted by the element radius and with the element centers as the tessellation generator points. This tessellation results in elements that

are contained completely within their polygons for element distributions that do not overlap (see Figure 1). We take the initial effective area for an element as the area of the radical Voronoi polygon associated with that element. Since the radical Voronoi tessellation covers the whole domain, the effective element area can represent 100% ice cover. The radical Voronoi polygon associated with an element is carried with it as the element moves.

Over time sea-ice deformation and ridging reduces sea-ice area (while increasing sea-ice thickness and approximately con-

serving sea-ice volume). One way to represent this in a DEM sea ice model is to allow elements to overlap as they ridge, and to concomitantly reduce the effective element area as they do. The radical Voronoi polygon associated with an element, $P_i$, can then be handled in either of two ways. Firstly, the size of the polygon associated with the element, $A_{P_i}$, can be kept constant. In this case $e < A_{P_i}$ in general between remappings. Secondly, as the effective element area of an element decreases during ridging the polygon can be decreased in size so that its area remains equal to the effective element area, i.e. $e = A_{P_i}$ between

remappings. The remapping method described here will work with either methodology. We will examine ridging in DEM sea ice models in a later work, but require the remapping scheme described here to periodically remap the element distribution back



to the initial Voronoi tessellation to ameliorate the effect of the element overlap associated with ridging during long duration simulations.

Sea-ice models used in current global climate models use numerous interdependent tracer fields that form a complex hierarchy, with ice concentration, $c$, or fractional area of ice in an element, as the root tracer. Sea ice models typically employ an ice thickness category distribution (e.g. Hunke et al., 2015; Rousset et al., 2015), where grid cell ice area is divided into a number of categories, each representing sea ice of different thicknesses. The sum of fractional areas in each thickness category is the total element ice concentration. For ice concentration less than 100%, the remaining area is assumed to be open water. Within each category the ice is further divided into vertical layers each of which contain tracer fields. Considering both categories and layers, MPAS-Seaice, for example, utilizes 23 different tracer fields for a typical physics simulation without biogeochemistry. Biogeochemistry uses many more additional tracers. Any remapping method must remap this complex tracer hierarchy in a computationally efficient manner, as well as ensuring that the sum of ice concentrations across thickness categories per element after remapping is bounded between 0 and 1.

Another desirable property of the remapping scheme is conservation of the appropriate conserved quantities. The effective area, $e$, provides a means to define conserved quantities that is consistent for a representation of sea ice with 100% ice concentration. For example, given ice concentration, $c$, ice thickness, $h$, and ice enthalpy, $q$, in element $i$ quantities that must be conserved during remapping are the total ice area per ice thickness category,

$$\mathcal{A}_k = \sum_i e_i c_{ik}, \tag{1}$$

the total ice volume per ice thickness category

$$\mathcal{V}_k = \sum_i e_i c_{ik} h_{ik}, \tag{2}$$

and the total ice energy per ice thickness category and ice layer

$$\mathcal{Q}_{kl} = \sum_i e_i c_{ik} h_{ik} q_{ikl}, \tag{3}$$

where each thickness category within an element is labelled by the $k$ index, and ice layers within a thickness category by the $l$ index. In the remapping implementation we distinguish between primitive tracer variables, such as thickness and enthalpy, and conserved quantities, such as volume and energy.

## 3 Geometric remapping implementation

Several properties are desirable in tracer remapping schemes:

- *Conservation:* A remapping scheme should preserve conservation of the appropriate quantities. Conservation of mass and energy are very important for long-term global climate simulations.





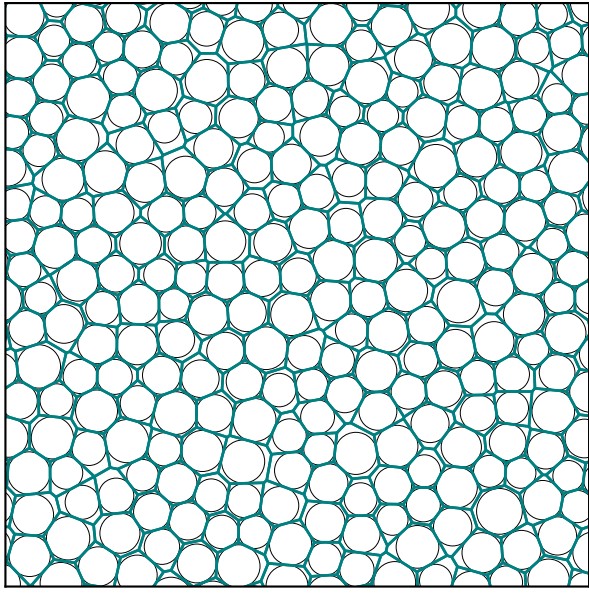

**Figure 1.** Example of an element distribution generated with the algorithm of Liang et al. (2015) showing the circular elements (*grey*), and the radical Voronoi tessellation (*green*).

– *Accuracy:* Since first order schemes are very numerically diffusive, the method should be at least second order accurate in space.

     – *Monotonicity:* The method should preserve monotonicity of the tracer fields so no new extrema are generated in these fields by remapping.

     – *Compatibility:* The method should ensure that no new extrema are created when the primitive tracer field is diagnosed
from the remapped conserved field, that is, both the primitive and conserved variables are consistently bounds-preserving.

     – *Computational efficiency:* The method should be computationally efficient for many tracers.

The geometric based remapping scheme proposed here satisfies all these requirements. The method proposed is based partly on the incremental remapping transport algorithm used for sea-ice transport in CICE and MPAS-Seaice (Dukowicz and Baumgardner, 2000; Lipscomb and Hunke, 2001). The scheme for remapping a source element distribution (indexed by $i$) to a
destination element distribution (indexed with $j$) is described below. This remapping method is used within a sea ice simulation to periodically remap a deformed sea ice element (source) distribution to an undeformed (destination) distribution, after which the simulation is restarted. Source element $i$ has its effective area polygon, $P_i$, advecting with it, while the fixed destination element $j$ has its effective area polygon, $P_j$, associated with it. Both polygons are convex. The remapping method is based on the polygon, $P_{ij}$, associated with the geometric overlap between the $P_i$ and $P_j$ polygons, which is also convex, and consists
of transferring quantities associated with $P_{ij}$ to $P_j$. Hence, a conserved quantity $Z_i$ on the source polygons is remapped to $P_j$





as

$$Z_j = \sum_i Z_{ij} \tag{4}$$

where $Z_{ij}$ is the part of $Z_i$ in the intersection polygon $P_{ij}$ and the sum is over the source element polygons, $P_i$, that overlap the destination polygon, $P_j$. Since the destination polygons tessellate the whole domain every part of all the $Z_i$ values, represented in the $Z_{ij}$ values, are transferred to a destination polygon ensuring conservation of $Z$.

For a first order method, where a tracer, $t_i$, is assumed constant within the $P_i$ polygons,

$$Z_{ij} = t_i e_{ij} \tag{5}$$

where $e_{ij}$ is the effective area associated with the overlap polygon $P_{ij}$ and is given by the fractional area of the overlap as

$$e_{ij} = e_i \frac{A_{P_{ij}}}{A_{P_i}} \tag{6}$$

where $A_{P_i}$ denotes the area of the $P_i$ polygon, and recall that $e_i$ is the effective area of element $i$ based on tessellation of the initial element configuration. For a second order method, with the tracer $t_i(\mathbf{x})$ represented as a linear function of position,

$$Z_{ij} = \frac{e_{ij}}{A_{P_{ij}}} \int_{A_{P_{ij}}} t_i(\mathbf{x}) \, dA. \tag{7}$$

The set of source elements does not necessarily fill the entire domain, as some portions of the domain can be made up of open water. The set of destination polygons, however, form a complete tessellation of the domain without gaps or overlap, since for conservation every part of the $P_i$ polygons must overlap with a $P_j$ polygon exactly once.

The major steps of the remapping method are described below:

1. Determine overlap polygons, $P_{ij}$, between the source ($P_i$) and destination ($P_j$) polygons and remap effective element area.

2. Compute linear reconstructions of average tracer fields on source elements based on tracer values in neighboring elements. The gradients in the reconstruction are limited to ensure monotonicity of the remapped fields in the case of uniform motion.

3. Integrate the conserved variables over the intersection polygons using the linear reconstructed tracer fields and aggregate in the destination polygons.

4. Enforce bounds preservation for remapped effective area and tracers using an optimization-based flux correction.

These steps are described in more detail in the following sections.





### 3.1 Polygon intersections and remapped area

In the first step of the algorithm, the intersection polygons, $P_{ij}$, are calculated using the algorithm of Preparata and Shamos (1985). In order to avoid the large computational cost of calculating the intersection polygon for every source and destination polygon pair, only destination polygons that are close enough to source polygons to have potential overlap are included in the

search. This is implemented with a link-cell method (Hockney et al., 1974; Plimpton, 1995).

Unlike for the discrete element application described here, when computing intersections between two well-defined grids that cover the same domain, the sum of the intersections will be equal to the domain area. Each destination grid cell will also be entirely covered by source grid cells so that the sum of the source intersections of a destination grid cell will equal the destination grid cell area. For our discrete element application, if the motion leading to the deformed element distribution is

non-uniform there may be gaps and overlaps between the source polygons used to compute the intersections. Additionally, near the ice edge there will be destination polygons that are only partially covered by source polygons. To account for this, we compute a remapped effective area, $e_j$, given by

$$e_j = \sum_i e_{ij} = \sum_i \frac{A_{P_{ij}}}{A_{P_i}} e_i. \tag{8}$$

Recall that $e_i$ may not be equivalent to the effective polygon area associated with element $i$ due to area changes from ridging.

The remapped effective area $e_j$ will, in general, not equal the destination element effective area $A_{P_j}$ and in some cases may exceed the destination element effective area. The optimization-based flux algorithm described in Section 3.4 is used to correct the remap effective area to ensure $e_j \leq A_{P_j}$ and correct tracer values with the computed area fluxes to enforce monotonicity while maintaining conservation. In the case of uniform motion there are no area changes due to ridging and the remapped effective area simplifies such that $e_i$ is equal to $A_{P_i}$ and $e_j$ will only differ from $A_{P_j}$ along the ice edge.

### 3.2 Linear tracer reconstruction

As the first step in remapping the tracers, a mean-preserving linear reconstruction of the tracer fields, $t_p(\mathbf{r})$, is made in each polygon of the source element distribution. For the concentration, $c$, thickness, $h$, and enthalpy, $q$, the reconstructions are

$$c_p(\mathbf{r}) = c + \alpha_c \nabla c \cdot (\mathbf{r} - \bar{\mathbf{r}}), \tag{9}$$

$$h_p(\mathbf{r}) = h + \alpha_h \nabla h \cdot (\mathbf{r} - \widetilde{\mathbf{r}}), \tag{10}$$

and

$$q_p(\mathbf{r}) = q + \alpha_q \nabla q \cdot (\mathbf{r} - \hat{\mathbf{r}}) \tag{11}$$

where $\mathbf{r} = (x, y)$ is the position vector within the element polygon, $\nabla c$, $\nabla h$, and $\nabla q$ are estimates of the tracer gradients for the $c$, $h$, and $q$ tracers in the source polygons, and $\alpha_c$, $\alpha_h$, and $\alpha_q$ are limiting coefficients for the $c$, $h$, and $q$ tracers that enforce





monotonicity. A linear tracer reconstruction ensures second-order spatial accuracy of the remapping. To satisfy conservation of $\mathcal{A}$, $\mathcal{V}$, and $\mathcal{Q}$, the reconstructed tracer fields must equal the known cell-averaged tracer values ($c$, $h$, and $q$) when integrated over the source polygons so that

$$e\frac{1}{A}\int_A c_p(\mathbf{r})\,dA = ce, \tag{12}$$

$$e\frac{1}{A}\int_A c_p(\mathbf{r})h_p(\mathbf{r})\,dA = che, \tag{13}$$

and

$$e\frac{1}{A}\int_A c_p(\mathbf{r})h_p(\mathbf{r})q_p(\mathbf{r})\,dA = chqe. \tag{14}$$

This requires that (Lipscomb and Hunke, 2001)

$$\bar{\mathbf{r}} = \frac{1}{A}\int_A \mathbf{r}\,dA, \tag{15}$$

$$\widetilde{\mathbf{r}} = \frac{1}{cA}\int_A c_p(\mathbf{r})\mathbf{r}\,dA, \tag{16}$$

and

$$\hat{\mathbf{r}} = \frac{1}{chA}\int_A c_p(\mathbf{r})h_p(\mathbf{r})\mathbf{r}\,dA. \tag{17}$$

Tracer gradients for a source element are calculated as a multivariate linear regression of the tracer values in that element and the neighboring source elements. For the $n_m$ neighbouring elements (including the element itself) the tracer gradients are given by

$$\nabla_x t = \frac{(YY)(XT) - (XY)(YT)}{(XX)(YY) - (XY)^2}, \tag{18}$$

and

$$\nabla_y t = \frac{(XX)(YT) - (XY)(XT)}{(XX)(YY) - (XY)^2} \tag{19}$$

where

$$XX = \sum_m x_m x_m - \frac{1}{n_m}\sum_m x_m \sum_m x_m, \tag{20}$$

$$YY = \sum_m y_i y_m - \frac{1}{n_m}\sum_m y_m \sum_m y_m, \tag{21}$$





$$XY = \sum_m x_m y_m - \frac{1}{n_m} \sum_m x_m \sum_m y_m, \tag{22}$$

$$XT = \sum_m x_m t_m - \frac{1}{n_m} \sum_m x_m \sum_m t_m, \tag{23}$$

$$YT = \sum_m y_m t_m - \frac{1}{n_m} \sum_m y_m \sum_m t_m, \tag{24}$$

where neighboring element $m$ has position $(x_m, y_m)$ and tracer value $t_m$. If two source polygons jointly overlap with a particular destination polygon they are defined as being neighboring source elements.

Tracer gradients are limited with a form of van Leer limiting (van Leer, 1979) to preserve monotonicity of the tracer fields. Tracer gradients are limited so that the extrema values of the linear reconstructed tracer field, $t_p(\mathbf{r})$, within a source polygon are within the range of tracers values for the surrounding neighbor source elements. These neighbor elements are defined in the same way as for the gradient calculation above. Since the extremal values of a linear reconstructed tracer field for a polygon are at the corners of that polygon, we use the minimum and maximum of the corner polygon tracer values of the reconstructed field, $t_i^{cmin}$ and $t_i^{cmax}$, to perform the limiting. The gradients for a source element $i$ are limited by multiplying them by a limiting coefficient, $\alpha_i$, which lies in the range [0,1] and whose value is given by

$$\alpha_i = \min(1, \alpha_{min}, \alpha_{max}) \tag{25}$$

where

$$\alpha_{min} = \begin{cases} \max\left(0, \dfrac{t_i^{min} - t_i}{t_i^{cmin} - t_i}\right), & \text{if } |t_i^{cmin} - t_i| > |t_i^{min} - t_i| \\ 1, & \text{otherwise} \end{cases} \tag{26}$$

and

$$\alpha_{max} = \begin{cases} \max\left(0, \dfrac{t_i^{max} - t_i}{t_i^{cmax} - t_i}\right), & \text{if } |t_i^{cmax} - t_i| > |t_i^{max} - t_i| \\ 1, & \text{otherwise} \end{cases} \tag{27}$$

where $t_i^{min}$ and $t_i^{max}$ are the minimum and maximum, respectively, of the tracer values of the surrounding neighbor source elements to source element $i$. This method restricts reconstructed tracer values of a given destination element, $t_j$, to be within the range of source tracer values for source elements that overlap with the destination element as well as source element neighbours of those source elements, as in the case for the incremental remapping algorithm. In addition, in a similar way gradients of the concentration fields for all the ice thickness categories are limited so that the sum of the reconstructed concentration over thickness categories lies within [0,1].




### 3.3 Integration over intersection polygons

With tracer gradients calculated and limited, conserved quantities can now be calculated from integrals of the reconstructed tracer fields across the intersection polygons. For destination polygon $j$

$$\mathcal{A}_{jk} = \sum_{ij} e_{ij} \frac{1}{A_{P_{ij}}} \int_{A_{P_{ij}}} c_p(\mathbf{r})\, dA, \tag{28}$$

$$\mathcal{V}_{jk} = \sum_{ij} e_{ij} \frac{1}{A_{P_{ij}}} \int_{A_{P_{ij}}} c_p(\mathbf{r}) h_p(\mathbf{r})\, dA, \tag{29}$$

and

$$\mathcal{Q}_{jkl} = \sum_{ij} e_{ij} \frac{1}{A_{P_{ij}}} \int_{A_{P_{ij}}} c_p(\mathbf{r}) h_p(\mathbf{r}) q_p(\mathbf{r})\, dA, \tag{30}$$

where the sum is over all intersections between destination element $j$ and source element distribution $i$, and the integral is over the $ij$ intersection polygon area. Finally, new remapped tracer values for the destination elements are determined with

$$c_{jk} = \frac{\mathcal{A}_{jk}}{e_j}, \tag{31}$$

$$h_{jk} = \frac{\mathcal{V}_{jk}}{e_j c_{jk}}, \tag{32}$$

and

$$q_{jkl} = \frac{\mathcal{Q}_{jkl}}{e_j c_{jk} h_{jk}}. \tag{33}$$

Only destination elements with a remapped effective area greater than zero then need to be kept for the continuing DEM simulation. To ensure consistency for partially ice filled destination elements before the final calculation of tracer values in equations 31 to 33, the effective area is set to the polygon area $A_{p_j}$, decreasing the concentration of sea ice calculated in equation 31.

Instead of directly calculating the integrals over source and intersection polygon area required by the remapping method, the integrands of these integrals are expanded so that only integrals of geometric quantities of the source polygons are required. These quantities are calculated once for all tracers improving computational efficiency for simulations with large tracer number.





For example equation 16 once expanded becomes

$$
265 \quad \widetilde{\mathbf{r}}_x = \frac{1}{cA} \left[ (c - \alpha_c \nabla c \cdot \bar{\mathbf{r}}) \int_A x \, dA + \right.
$$

$$
(\alpha_c \nabla_x c) \int_A x^2 \, dA +
$$

$$
\left. (\alpha_c \nabla_y c) \int_A xy \, dA \right], \tag{34}
$$

and

$$
\widetilde{\mathbf{r}}_y = \frac{1}{cA} \left[ (c - \alpha_c \nabla c \cdot \bar{\mathbf{r}}) \int_A y \, dA + \right.
$$

$$
270 \qquad (\alpha_c \nabla_x c) \int_A xy \, dA +
$$

$$
\left. (\alpha_c \nabla_y c) \int_A y^2 \, dA \right] \tag{35}
$$

and only source polygon integrals of $x$, $y$, $x^2$, $y^2$, and $xy$ are required. The integrals are calculated by breaking polygons into triangles by connecting the polygon centroid to the polygon vertices and using the Gaussian triangular quadrature rules of Dunavant (1985).

275 ## 3.4 Optimization-based flux correction to remapping

For uniform motion of the elements there is no relative motion (either rectilinear or rotational) between elements, so their polygons do not overlap during their motion. For general motion, however, relative motion between elements results in element polygons overlapping. For the formulation of effective area from Section 2, this would potentially result in the remapped effective area exceeding the available geometric area of the destination polygons, resulting in remapped concentrations greater
280 than 1. To ameliorate this issue we implement a flux based optimization correction to the effective area and tracer fields that is applied after the remapping discussed above. This optimization scheme determines a minimal flux, $f_{j_1 j_2}$, between polygons $j_1$ and $j_2$ of the destination element distribution that conservatively removes excess effective area from destination polygons, so after the flux is applied to the remapped effective area and tracer fields the effective areas, $e_{j_1}$ and $e_{j_2}$, are less than the destination polygon areas, $A_{P_{j_1}}$ and $A_{P_{j_2}}$ for all destination polygons. For a given destination polygon $P_{j_1}$ we require that the
285 corrected effective area, $e'_{j_1}$, obey

$$
0 \leq e'_{j_1} \leq A_{P_{j_1}}. \tag{36}
$$

The corrected effective area is given by

$$
e'_{j_1} = e_{j_1} + \sum_{j_2} B_{j_1 j_2} f_{j_1 j_2} \tag{37}
$$





where $B_{j_1 j_2}$ is a sparse matrix with entries of either 0 (if polygons $j_1$ and $j_2$ do not share an edge), -1, or 1 (signifying whether
the flux $f_{j_1 j_2}$ is into or out of the polygon $j_1$). Thus the determination of the set of fluxes, $f_{j_1}$, can be cast in the following
inequality constrained quadratic program:

$$
\begin{aligned}
\text{minimize} \quad & \tfrac{1}{2}\mathbf{f}^T \mathbf{P} \mathbf{f} \\
\text{subject to} \quad & \mathbf{l} \le \mathbf{B}\mathbf{f} \le \mathbf{u}
\end{aligned}
\tag{38}
$$

where $\mathbf{f}$ is the vector of edge fluxes, $f_{i_1 j_2}$, $\mathbf{P}$ is the identity matrix, $\mathbf{B}$ is the matrix $B_{j_1 j_2}$, and $\mathbf{u}$ and $\mathbf{l}$ are vectors defined by

$$
l_j = -e_j
\tag{39}
$$

and

$$
u_j = A_{P_j} - e_j.
\tag{40}
$$

The quadratic program is solved in parallel by the PermonQP library (Hapla et al., 2016; Kružík et al., 2020; Hapla et al.,
2021), which uses the PETSc library (Balay et al., 1997, 2019). The fluxes, $f_{j_1 j_2}$, are then used to conservatively transfer
effective area and tracer quantities between destination polygons, removing excess effective area. Since the fluxes are fluxes of
effective area, a conserved quantity, $Z$, is corrected according to

$$
Z'_{j_1} = Z_{j_1} - Z_{j_1} \frac{f_{j_1 j_2}}{e_j},
\tag{41}
$$

and

$$
Z'_{j_2} = Z_{j_2} + Z_{j_1} \frac{f_{j_1 j_2}}{e_j}
\tag{42}
$$

for all values of $\mathbf{f}$, where $f_{j_1 j_2}$ flows from element $j_1$ to $j_2$.

Another issue ameliorated by the flux correction is the overlapping of source elements with coastal elements. Here we
represent coastlines through a series of immovable elements coincident with land. DEM models, however, model compressive
interaction through an overlap of elements, so sea ice elements pushed by winds onto fixed coastal elements will overlap
slightly with those elements. During the remapping described previously some sea ice will be remapped onto the coastal
elements. This can be fixed by moving this remapped ice to the nearest destination element that is not a coastal element before
the flux correction is performed. While moving this ice has the potential to increase the effective area of these elements to be
larger than their geometric area, the flux correction algorithm ameliorates this effect.

### 3.5    Effect of open water

One difference between particle and Eulerian methods that causes potential issues for remapping is the treatment of open water.
In Eulerian methods, empty Eulerian cells, representing open water, can be included in the calculation of tracer gradients and
monotonicity limits. For DEM sea ice models, elements only exist where sea ice exists so there are no empty elements,
representing open water, to be included in tracer gradient calculations, or in the neighboring element tracer extrema used to





limit those gradients. A possible effect of this is that tracer values for ice concentration and thickness, $t_i$, for elements at the ice pack edge are likely to be the minimum of the neighbour tracer values, so that $t_i = t_{ij}^{min}$. From equations 25 to 27 this would mean the gradients would be limited to zero in these ice edge elements, and the remapping here would revert to a highly

diffusive first order method. To investigate the effect of this issue we implement a method to account for open water in the tracer gradient calculation and limiting. Firstly, for the neighboring source elements, $N_i$, of a given source element (not including the element itself) we calculate the total effective area, $E_i$, and the centroid of that effective area, $\mathbf{r}_{E_i}$. These are given by

$$E_i = \sum_{i}^{N_i} e_i \tag{43}$$

and

$$\mathbf{r}_{E_i} = \frac{1}{E} \sum_{i}^{N_i} e_i \mathbf{r}_i. \tag{44}$$

If the $E_i$ is below some limit we assume that there is some open water surrounding the source element and include an additional open water neighbor pseudo-element into the gradient and gradient limiting calculations. This pseudo-element has zero concentration, and a position on the opposite side of the element to the centroid of the surrounding effective area $(2\mathbf{r}_i - \mathbf{r}_{E_i})$. The addition of a neighbor concentration value of zero would prevent the gradient at the ice edge from being limited to zero.

We include the zero concentration pseudo-element if the sea ice surrounding the element of interest is below a fraction (taken as 0.75) of an estimate of the expected value of $E_i$ for a fully covered pack, $E_i'$. In two dimensions we estimate $E_i'$ based on a hexagonal close packing of elements, so that $E_i' \simeq 6\bar{A}_{P_j}$, where $\bar{A}_{P_j}$ is the mean area of destination polygons. In one dimension each element is only surrounded by two elements so for one dimensional tests we use $E_i' \simeq 2\bar{A}_{P_j}$.

## 4 Computational results

We now test the remapping algorithm described in the previous section with a series of idealized test cases. Motion of the elements between remapping is determined by the LAMMPS (Plimpton, 1995) molecular dynamics code, which has the capability to run DEM simulations, wherein particle interaction forces are computed based on contacts, and the equations of motions are integrated in time. LAMMPS can either explicitly enforce the motion of elements (such as in uniform rectilinear motion), or solve the equations of motion of the elements with a modified velocity Verlet integrator (Swope et al., 1982). For

test cases involving solving the equations of motion, we solve a simplified sea-ice momentum equation

$$m \frac{\partial \mathbf{u}}{\partial t} = \mathbf{F} + \tau_a + \tau_w. \tag{45}$$

In a DEM context, this reduces to a set of coupled first-order equations for all elements:

$$m \frac{d\mathbf{u}_i}{dt} = \mathbf{F}_i + \tau_a(\mathbf{x}_i) + \tau_w(\mathbf{x}_i) \tag{46}$$

where $m_i$ is the mass of each element, $\mathbf{u}_i$ is the element velocity, $\mathbf{F}_i$ is the sum of contact forces on element $i$, and $\tau_a$ and $\tau_w$

are the atmospheric and ocean stresses, respectively, evaluated at the location of the element $\mathbf{x}_i$. These surface stresses have a





quadratic form given by

$$\tau = ec\rho|\mathbf{U} - \mathbf{u}|(\mathbf{U} - \mathbf{u}) \tag{47}$$

where $e$ is the effective element area, $c$ is a drag coefficient, $\rho$ is the fluid density (air or water), and $\mathbf{U}$ is the fluid velocity (atmospheric wind or ocean current). This momentum equation is discretized in time as

$$
\begin{aligned}
\quad m\frac{\mathbf{u}^n - \mathbf{u}^{n-1}}{\Delta t} = \ &\mathbf{F}_{n-1} \\
&+ ec_a\rho_a\left|\mathbf{U}_a - \mathbf{u}^{n-1}\right|(\mathbf{U}_a - \mathbf{u}^{n-1}) \\
&+ ec_w\rho_w\left|\mathbf{U}_w - \mathbf{u}^{n-1}\right|(\mathbf{U}_w - \mathbf{u}^{n})
\end{aligned}
\tag{48}
$$

where superscripts $n$ and $n-1$ signify the current and previous time steps, respectively. The ocean velocity is treated partly implicitly for stability, and a simple Hookean elastic repulsion contact model is employed for element collision for the test cases

considered here. The air drag coefficient, $c_a$, and air density, $\rho_a$, are given by 0.0012 and $1.3\,\mathrm{kg\,m^{-3}}$, respectively (Hibler, 1979), while the ocean drag coefficient, $c_w$, and water density, $\rho_w$ are given by of 0.00536 and $1026\,\mathrm{kg\,m^{-3}}$ (Kim et al., 2006), respectively.

## 4.1 One dimensional uniform motion

The first test we perform is a simple one dimensional test case with uniform motion and a "top hat" initial ice concentration
distribution. The destination element distribution consists of uniform elements with radius, $r$, of 500 m arranged in the $x$ direction across a domain 1000 km long. The domain accommodates a single element in the $y$ direction and is 1 km wide. The initial source distribution is a subset of the destination distribution consisting of the 100 elements between 100 km and 200 km in the $x$ direction. Initial ice concentration, $c_0(x)$, is given by

$$c_0(x) = \begin{cases} 1, & \text{for } x_1 \le x \le x_2 \\ 0, & \text{otherwise.} \end{cases} \tag{49}$$

where $x_1$ and $x_2$ are 100 km and 200 km, respectively. These elements have an initial effective area of a square of side $2r$, a sea-ice concentration of 1 and a sea-ice thickness of 1 m. The elements undergo uniform motion of $500\,\mathrm{m\,s^{-1}}$ in the positive $x$ direction, so that after 200 seconds the distribution has moved 100 km, whereupon the final source distribution is examined. Figure 2 shows this final distribution for three situations. The first situation ("No remap" in Figure 2) has no remapping performed during the uniform motion so that the final distribution of ice concentration and thickness is the same as the initial

one, other than being translated 100 km in the positive $x$ direction. For the second situation ("Low order" in Figure 2) the source distribution undergoes remapping every second for a total of 200 remappings. With these values the elements move half their diameter before remapping. Tracer gradients are set to zero so that the remapping algorithm becomes a low order method. The final situation ("Higher order" in Figure 2) is the same as the "Low order" situation except tracer gradients are not set to zero, making the remapping a more accurate higher order method. The "No remap" situation is useful for demonstrating the final

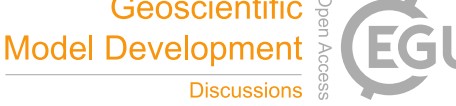

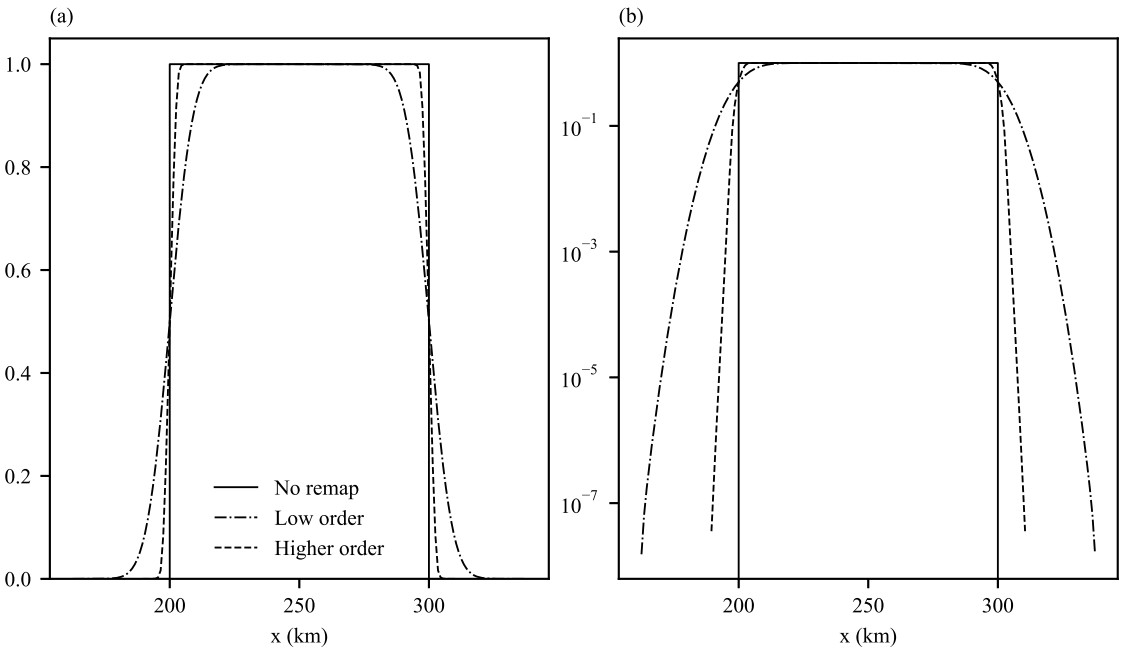

**Figure 2.** (a) Final ice concentration for the one dimensional top hat test case. (*solid*): No remapping, (*dash-dot*): Low order remapping, (*dashed*): Higher order remapping. (b) As (a) but with a logarithmic ordinate axis. Results for no remapping, low order remapping, and higher order remapping are shown.

distribution that a perfect remapping scheme, with no numerical diffusion, would produce. Figure 2 shows that the "Higher order" method produces significantly less numerical diffusion of the concentration profile than the "Low order" method.

We now use this test case to explore the effect of taking account of open water in the tracer gradient and limiting, as described in section 3.5. We rerun the "Higher order" simulation with and without the effects of open water taken into account (see Figure 3). As can be seen in Figure 3 the effect of the absence of open water elements appears negligible with only a

very minor amount of extra numerical diffusion present at the very edge of the ice due to this absence (see inset in Figure 3b). Similar results were found with the two dimensional simulations described next. Based on these results we neglect the effect of the absence of open water elements for the remainder of this work.

The next one dimensional test case is based on that found in Lipscomb and Hunke (2001), and is used to test how well the remapping algorithm preserves the compatibility of the particle tracers. The test case is the same as the previous one dimen-

sional test case, except for the initial distribution of ice concentration, thickness and volume. These initial distributions are more complex than the previous test case (see Figure 4a,b) and provide a harder test of compatibility. Initial ice concentration,





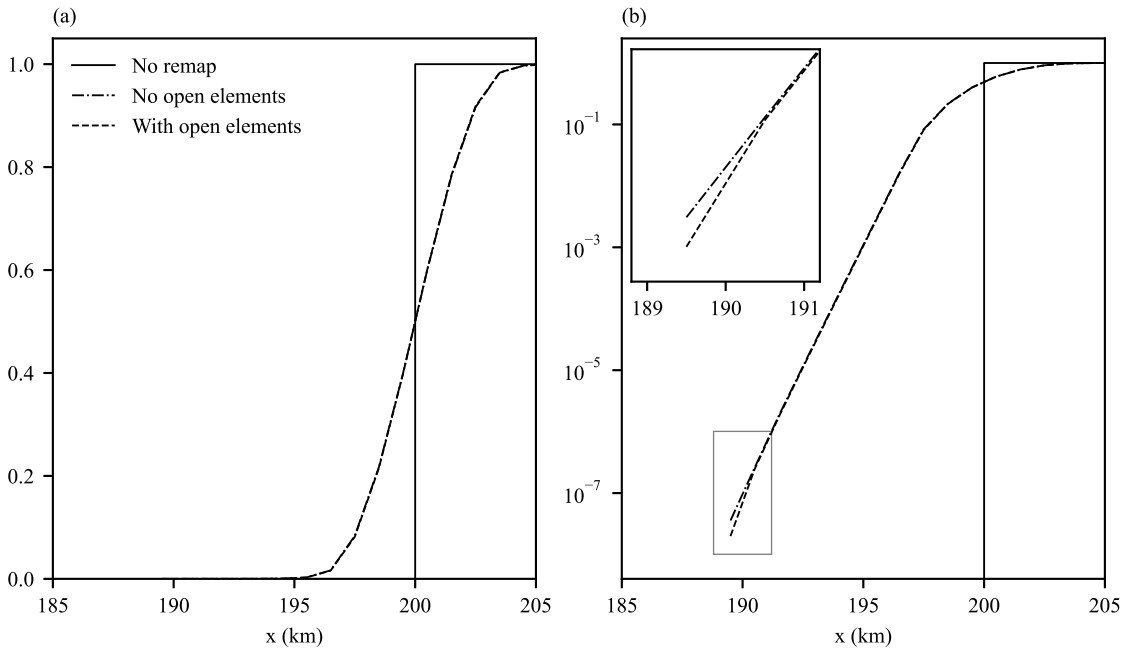

**Figure 3.** (a) Detail of ice concentration for the top hat test case showing the higher order remapping results with and without taking into account the effect of open water. (b) As (a) but with a logarithmic ordinate axis. The inset shows detail of the differences at the extreme of the distribution.

$c_0(x)$, and thickness, $h_0(x)$, are given by

$$c_0(x) = \begin{cases} \frac{x-x_1}{x_3-x_1}, & \text{for } x_1 \leq x \leq x_3 \\ 1, & \text{for } x_3 \leq x \leq x_5 \\ 0, & \text{otherwise.} \end{cases} \tag{50}$$

and

$$h_0(x) = \begin{cases} 0.25, & \text{for } x_2 \leq x \leq x_4 \\ 1, & \text{for } (x_1 \leq x < x_2) \textbf{ or } (x_4 < x \leq x_5) \end{cases} \tag{51}$$

where $x_1$, $x_2$, $x_3$, $x_4$, and $x_5$ are given by 100 km, 112.5 km, 150 km, 187.5 km, and 200 km respectively. As before we preform three separate simulations: One with no remapping (Figure 4a,b), one with low order remapping (Figure 4c,d), and one with higher order remapping (Figure 4e,f). As before both the low and higher order remapping schemes are capable of remapping the initial particle distribution, with the higher order scheme producing much less numerical diffusion. Figures 4d,f also clearly show that the thickness tracer preserves monotonicity as well as the concentration and volume, clearly demonstrating compatibility of the method.

**Figure 4.** One dimensional compatibility test case. (a, b): No remapping, (c, d): Low order remapping, (e, f): Higher order remapping. (a, c, e): Ice concentration, and volume, (b, d, f): Ice thickness.





## 4.2 Two dimensional uniform motion

The next test case we examine involves a two dimensional domain of size 300 km in the $x$ direction and 100 km size in the $y$ direction. The destination and initial source element distributions consist of a random arrangement of circular elements
with a distribution of element radius, with mean radius $\sim 513\,\mathrm{m}$. To create the tightly packed initial element distribution we use the method of Liang et al. (2015). This method uses a Lloyd (1982) like iteration of the radical Voronoi tessellation of an initial random set of generator points each with an assigned radius drawn from a distribution. During each iteration the generator locations are moved to the centers of the largest inscribed circles within the radical Voronoi polygons (see Figure 1 for an example element distribution generated with this method). We explore two different initial source element distributions
discussed in Lipscomb and Ringler (2005). The first is a cosine bell distribution with an initial ice concentration, $c_0(x, y)$, given by

$$c_0(x,y) = \begin{cases} \frac{1}{2}\left(1 + \cos\left(\frac{\pi r}{r_0}\right)\right), & r < r_0 \\ 0, & \text{otherwise.} \end{cases} \tag{52}$$

where $r = \sqrt{(x-x_0)^2 + (y-x_0)^2}$, $(x_0, y_0)$ is the center of the cosine bell at (25 km, 50 km), and $r_0$ is the radius of the cosine bell with size 15 km. The second is a slotted cylinder with an initial ice concentration, $c_0(x, y)$, given by

$$c_0(x, y) =$$

$$\begin{cases} 1, & (r < r_0)\ \textbf{and not} \\ & \left((|x - x_0| < \frac{1}{6}r_0)\ \textbf{and}\ (y - y_0) > -\frac{2}{3}r_0)\right) \\ 0, & \text{otherwise.} \end{cases} \tag{53}$$

where $x_0$, $y_0$, and $r_0$ are the same as for the cosine bell distribution. Elements on the edge of the slotted cylinder are assigned an initial concentration according to their areal fraction within the slotted cylinder. The element distributions are moved uniformly in the $x$ direction with speed $500\,\mathrm{m\,s^{-1}}$ and examined after 200 seconds. As before we perform three simulations, one with
no remapping, one with low order remapping and one with higher order remapping, with remapping performed every second for a total of 200 remappings. Spatial maps of the element concentration at the end of the simulation are shown in Figure 5. The higher order remapping scheme is able to preserve the approximate shape of the initial concentration distributions while high numerical diffusion with the low order remapping scheme leaves the final element distribution more spread out with any structure of the slotted cylinder lost. The difference in numerical diffusion of the two remapping methods is evident in cross
sections of final ice concentration through the middle of the distributions, as shown in Figure 6. As can be seen almost no trace of the slot in the slotted cylinder is present for the low order scheme, while the slot is preserved well for the higher order method.

Next we study the error scaling properties of the remapping method. To do this we create various initial element distributions with different mean particle radii, but the same relative distribution of radii. We rerun the simulations for these element

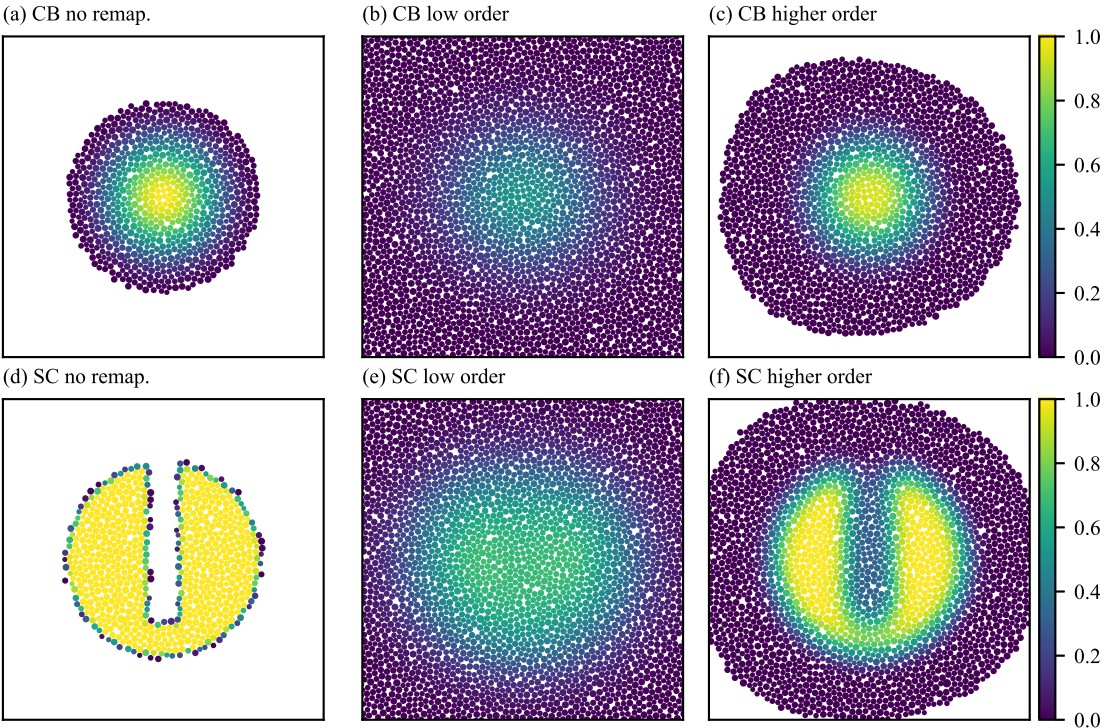

**Figure 5.** Final ice concentration for the two dimensional remapping test case with an initial cosine bell (CB: a, b, c) and slotted cylinder (SC: d, e, f) particle distribution. (a, d): No remapping, (b, e): Low order remapping, (c, f): Higher order remapping.

distributions and record the $L_2$ error norm for each simulation. We calculate the $L_2$ error norm as

$$L_2 = \frac{\sum_i e_i \sqrt{c_i - c_i^0}}{\sum_i e_i \sqrt{c_i^0}} \tag{54}$$

where the sum is over the final element distribution, $e_i$ and $c_i$ are the effective area and concentration of the final distribution and $c_i^0$ is the ice concentration for no remapping. In Figure 7, the values of the $L_2$ error norm are plotted against the mean radius of the element distribution for both the cosine bell and slotted cylinder initial distributions and the low and higher order remapping methods. As can be seen from the figure the higher order method has significantly lower errors than the low order method for both initial distributions. The gradient of the slope for the smooth cosine bell distribution is also much higher than the slotted cylinder with its sharp discontinuities. It is also apparent that for the smooth cosine bell distribution the remapping method is approximately second order accurate for the higher order method.

## 4.3 Flux correction tests

Having demonstrated in sections 4.1 and 4.2 that the higher order remapping method for uniform motion is capable of remapping the particle distribution with low numerical diffusion while preserving compatibility, we next turn to examining the effect



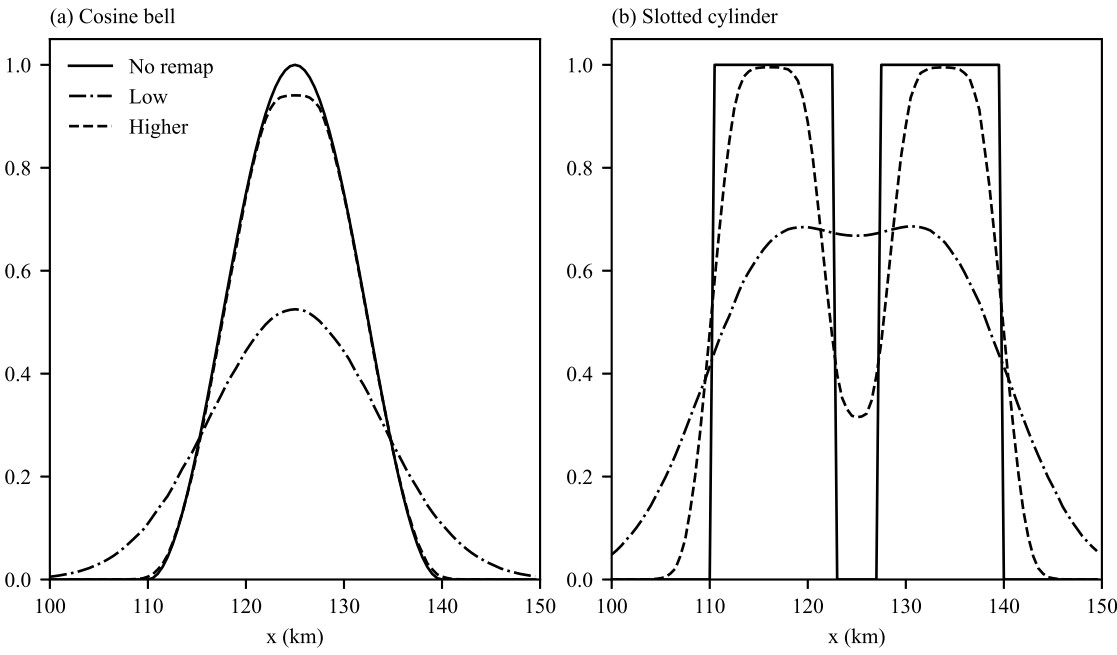

**Figure 6.** Cross section ($y = 50\,\mathrm{km}$) of ice concentration for the two dimensional test case for the initial (a) cosine bell, and (b) slotted cylinder particle distributions. Results for no remapping, low order remapping, and higher order remapping are shown.

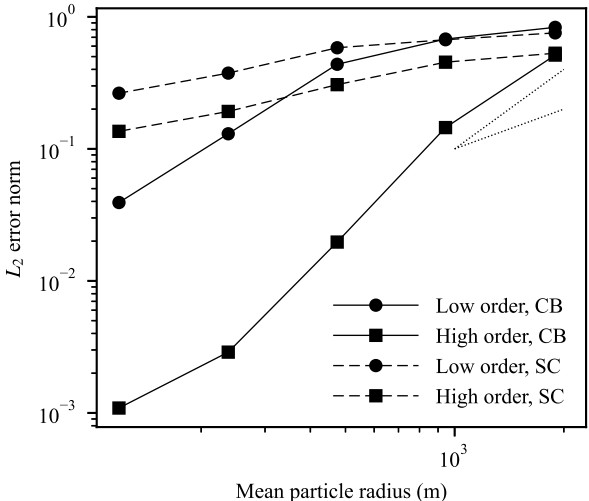

**Figure 7.** $L_2$ error norm versus mean particle radius for the cosine bell (CB) and slotted cylinder (SC) two dimensional test cases with both low and higher order remapping. First and second order convergence gradients are also shown as the dotted lines.





of non-uniform motion and testing the ability of the flux correction algorithm described in section 3.4 to correct the issues created by this motion. The first test we examine is a variation of the two dimensional test used in the previous section. Instead of making the elements undergo uniform motion, we fix their positions and randomize the orientations of their associated

polygons before each remapping. This introduces the overlaps and gaps between source polygons that might be expected from a general motion of the elements. As before, a total of 200 remappings are performed before the concentration distributions are examined. Figures 8b and e show the effect of these randomized orientations on the remapped ice concentration for the higher order remapping scheme. As can be seen, while the general shape of the initial distributions is preserved, the randomization results in ice concentrations significantly larger than one for some elements (where elements overlapped) and ice concentration

less than that expected for others (from gaps in the polygon distribution), with these effects taking on a random nature. Clearly, without correction this effect would have a seriously detrimental effect for any sea-ice simulation. Figures 8c and f show the same final ice concentration distribution, but generated with the flux correction scheme described in section 3.4 proceeding remapping. Now, ice concentrations are bounded correctly by [0,1], and the final distribution is smooth with similar results to that obtained in section 4.2. This is also evident in the cross section of ice concentration shown in Figure 9. Repeating the $L_2$

error scaling analysis in the previous section (Figure 10) shows the larger $L_2$ error norms for the uncorrected simulations, while the flux corrected simulations show significantly lower errors, and also show second order convergence with mean particle size for the smooth cosine bell test case.

To demonstrate the effect of the flux correction in more realistic situations we present two more test cases. In the first, adapted from Lipscomb et al. (2007), we have a square domain of size 1000 km by 1000 km with a random distribution of elements

covering the domain generated by the algorithm of Liang et al. (2015). In the middle of the domain is an 'L' shaped island consisting of fixed coastal elements. Initial elements are coastal or sea-ice elements according to

**if** $(x > x_0$ **and** $x <= x_1$ **and** $y > y_0$ **and** $y <= y_1)$ **and not** $(x > x_0$ **and** $x <= x_2$ **and** $y > y_0$ **and** $y <= y_2)$ **then**

Coastal element

**else**

Sea-ice element

**end if**

where $(x, y)$ is the position of the element center, $x_0$ and $y_0$ equal 400 km, $x_1$ and $y_1$ equal 600 km, and $x_2$ and $y_2$ equal 550 km. Elements have a mean element radius of $\sim 10\,\mathrm{km}$, with non-coastal elements having a concentration of one everywhere. A constant wind forcing is applied to the elements with the $x$ and $y$ components of wind velocity both equal to $10\,\mathrm{m\,s^{-1}}$,

while the ocean is stationary. During the simulation the elements flow around the island leaving an open water wake behind it. The simulation is run for 2 days after which the element distribution is remapped to the destination element distribution with the higher order scheme and either with or without the flux correction algorithm. Figure 11a and b show the effect on ice concentration of remapping without and with the flux correction, respectively. Figure 11c and d show the same but with a value of one subtracted from the ice concentration to highlight the excess ice concentration generated by the remapping when the

flux correction is not used. These figures clearly show elements with concentration greater than one after remapping without





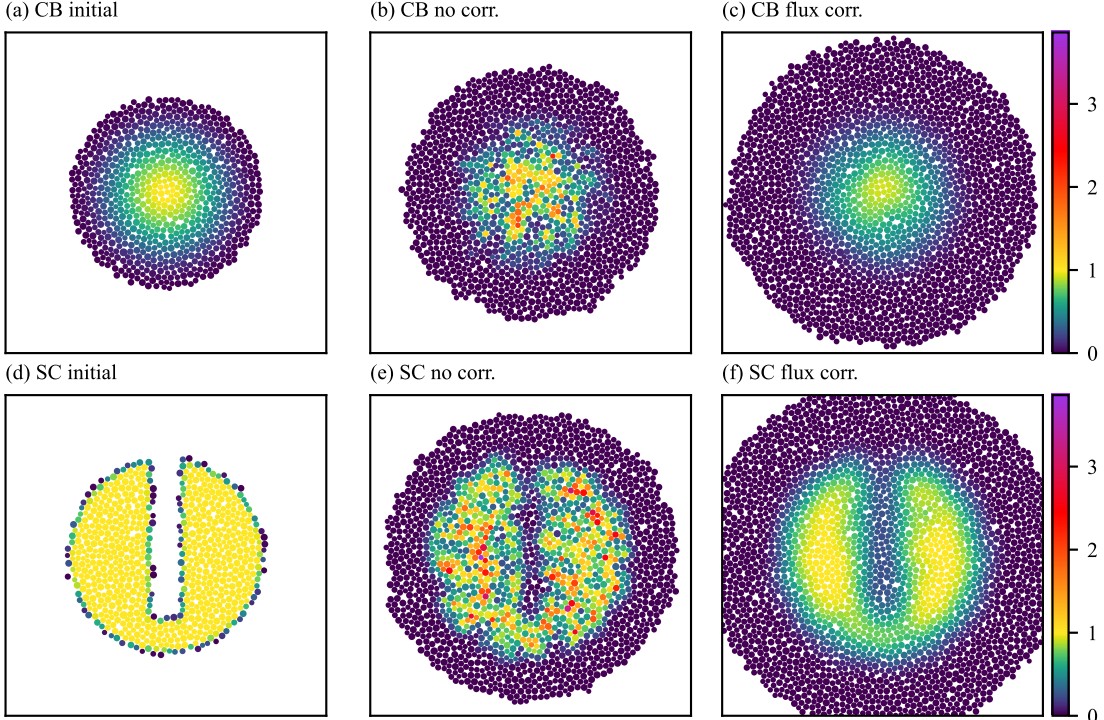

**Figure 8.** Stationary two dimensional remapping test case with randomization of element orientations for cosine bell (CB: a, b, c) and slotted cylinder (SC: d, e, f) initial distributions. Shown are the initial distributions (a and d), results without flux correction (b and e), and results with flux correction (c and f).

flux correction and also show how the flux correction removes this excess concentration and results in a concentration field closer to the concentration field before remapping with a concentration of one most places within the pack.

The final test case we examine is that from Flato (1993). Here the domain consists of a 500 km by 500 km closed square with the upper half ($y > 250$ km) initially filled with elements with a concentration of one and a mean radius of $\sim 2.5$ km. The
elements have the same air and ocean stress formulation and the same elastic repulsion contact model as the previous island test case. The ocean, again, is stationary but the wind field, $\mathbf{U}_a$, takes the form of a vortex with

$$\mathbf{U}_a(\mathbf{r}) = \min\left\{\omega|\mathbf{r}|, \frac{\lambda}{|\mathbf{r}|}\right\}\left(\mathbf{k} \times \frac{\mathbf{r}}{|\mathbf{r}|}\right) \tag{55}$$

where $\mathbf{r}$ is the position vector relative to the center of the vortex at (250 km, 200 km), $\omega$ is $0.5 \times 10^{-3}\,\mathrm{s^{-1}}$, $\lambda$ is $8 \times 10^5\,\mathrm{m^2\,s^{-1}}$, and $\mathbf{k}$ is the unit vertical vector. The simulation is run for 2 days and then the element distribution is remapped. As for the
island test case, Figure 12 shows that the flux correction scheme is capable of removing excess concentration caused by more realistic non-uniform motion of the elements for this test case.





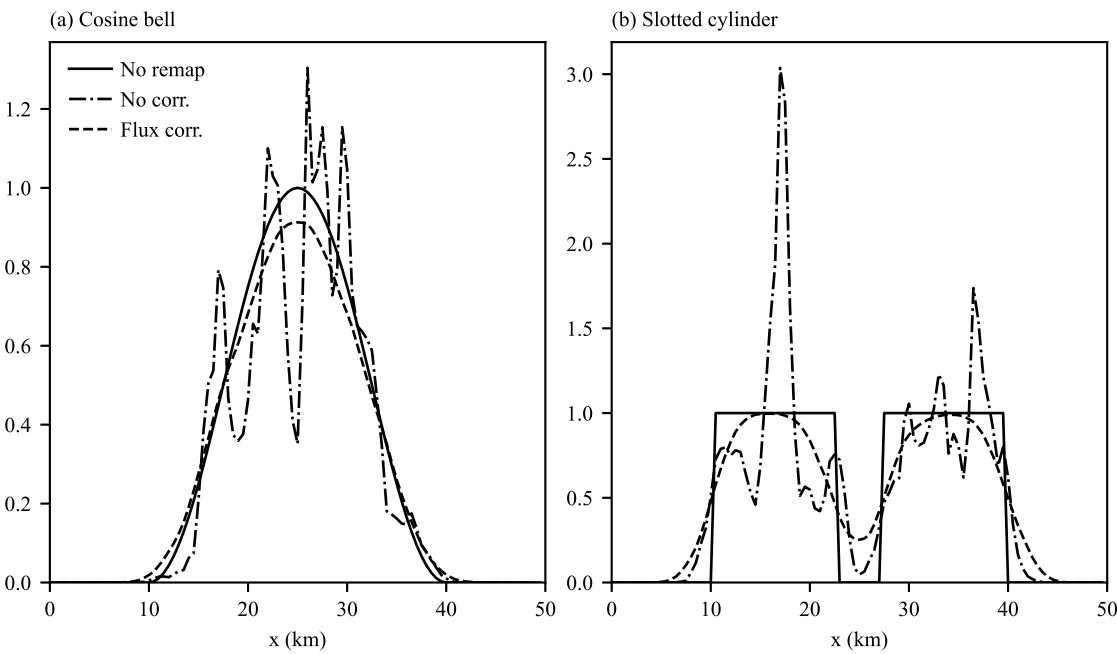

**Figure 9.** Cross section ($y = 50$ km) of ice concentration for the stationary two dimensional test case with randomized orientations for the initial (a) cosine bell, and (b) slotted cylinder particle distributions.

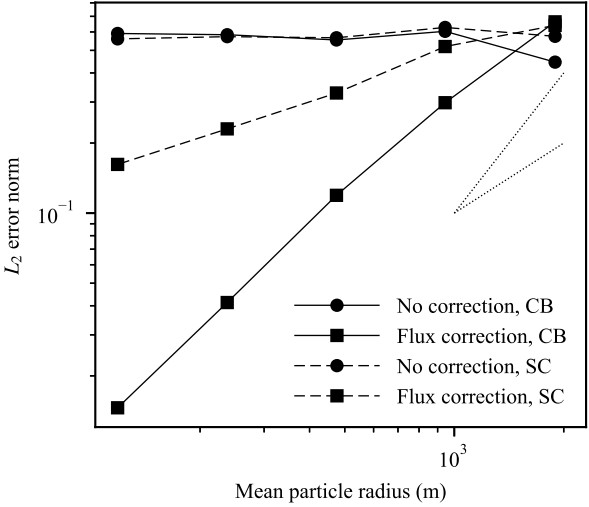

**Figure 10.** $L_2$ error norm versus mean particle radius for the cosine bell (CB) and slotted cylinder (SC) stationary two dimensional test cases with randomized orientations with both low and higher order remapping. First and second order convergence gradients are also shown as the dotted lines.

**Figure 11.** Island test case showing remapping without any correction (a, c) and with the flux based correction (b, d). (a, b) show ice concentration, while (c, d) show the difference from a concentration of 1.0. Coastal elements comprising the central island are shown in grey.

(a) No correction

(b) Flux-based correction

(c) No correction

(d) Flux-based correction

**Figure 12.** Vortex test case showing remapping without any correction (a, c) and with the flux based correction (b, d). (a, b) show ice concentration, while (c, d) show the difference from a concentration of 1.0.





# 5 Conclusions

Modeling sea ice dynamics with the discrete element method has the potential to enable the capture of the anisotropic deformation and fracture seen in observational data that is difficult to reproduce in continuum models. The DEM models, however,

present a challenge due to convergence and overlap of the Lagrangian elements in simulations. To manage this element clustering, a deformed element distribution is periodically remapped to an undeformed distribution. This paper presents a geometric-based remapping algorithm designed to address the unique challenges of accurately remapping sea ice tracer fields for circular discrete elements. In particular, the method includes a representation of effective element area defined with a radical Voronoi tessellation for the undeformed element configuration that enables an accurate definition of area conservation in the case of

100% sea ice concentration. Our remapping approach builds on conservative, bounds preserving, and compatible algorithms developed for incremental remapping and applied to sea ice (Lipscomb and Hunke (2001); Lipscomb and Ringler (2005)). In the case of uniform motion, the effective element area enables a complete coverage of the ice domain without overlap and the method inherits the properties of the incremental remapping algorithms. In the case of non-uniform motion, where gaps and overlaps occur between the deformed effective element areas, monotonicity can be lost. To address this, we developed a

novel flux-based correction method, which maintains conservation while correcting for bounds violations due to overlapping elements. Numerical examples are provided that demonstrate the second-order accuracy of the method, bounds preservation for both uniform and non-uniform motion, and compatibility between primitive and conserved tracer quantities.

*Code availability.* The code and test cases used in this paper can be found at http://doi.org/10.5281/zenodo.4918956.

*Author contributions.* AKT developed the methodology of the remapping scheme and the remapping software. AKT also implemented the

test cases and performed model validation and analysis. KJP contributed to the remapping methodology and software development, as well as model analysis. DB integrated the LAMMPS model into the remainder of the code. AKT and KJP wrote this manuscript with DB contributing to the editing of the manuscript. All authors contributed to funding acquisition.

*Competing interests.* The authors declare that they have no conflict of interest.

*Acknowledgements.* This material is based upon work supported by the U.S. Department of Energy, Office of Science, Office of Advanced

Scientific Computing Research and Office of Biological and Environmental Research, Scientific Discovery through Advanced Computing (SciDAC) program. The authors wish to thank the PermonQP team for help in using the PermonQP library, and Andrew Roberts for contributing to the writing of the SciDAC DEMSI proposals. Sandia National Laboratories is a multi-mission laboratory managed and operated by National Technology and Engineering Solutions of Sandia, LLC., a wholly owned subsidiary of Honeywell International, Inc., for the U.S.





Department of Energy's National Nuclear Security Administration under contract DE-NA0003525. This paper describes objective technical
results and analysis. Any subjective views or opinions that might be expressed in the paper do not necessarily represent the views of the U.S.
Department of Energy or the U.S. Government.





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
