# Peer review of "Geometric remapping of particle distributions in the Discrete Element Model for Sea Ice (DEMSI v0.0)"

_Geoscientific Model Development, 2021_

## Author Comment (AC1)

**Response to reviewers of "Geometric remapping of particle distributions in the Discrete Element Model for Sea Ice (DEMSI v0.0)"**

A. K Turner, K. J. Peterson, and D. Bolintineanu

December 6, 2021

We thank the reviewers for their comments and suggestions. We have replied to the comments below, with original reviewer comments in italics.

**1 Response to reviewer 1 comments**

**1.1 Specific Comments**

- *Line 205 mentions neighboring source elements. Are neighbors in this case defined as all of the source polygons that intersect with a single target polygon?*

    - Yes, this is how neighbor source elements are defined. For greater clarity, We have moved the line "If two source polygons jointly overlap with a particular destination polygon they are defined as being neighboring source elements." to earlier in the manuscript.

- *Is equation (28) correct? Should the sum be over i instead of ij? Also, ice thickness category k is listed on the left-hand side of the equation. Should it also be included on the right-hand side?*

    - We have changed the sum to be over i and made other subscript changes for clarity. We have also added the k and l indices to appropriate quantites on the right-hand side of equations 28-30.

- *Line 259 states that " the effective area is set to the polygon area Apj " Is this statement in conflict with equation (8)? Where the effective area of the target polygon is calculated.*

    - This is compatible as explained in the additional text added: "This is compatible with equation 8 since the values of effective area and concentration are not unique for representing the physically relevant quantity of sea ice area within an element, given by $A_i = c_i e_i$. Provided the area of sea ice within an element remains fixed and $0 \leq c_i \leq 1$ and $0 \leq e_i \leq A_p$, the concentration and effective area can vary together without affecting the physical representation."

- *Line 449 mentions "the final distribution is smooth with similar results to that obtained in section 4.2" Should it be exactly the same? And if so is it worth comparing the two results?*

    - These should not be exactly the same because the particles distribution prior to remapping in the first case is from uniform motion, while the second is from nonuniform motion resulting in gaps and overlaps between particle effective areas. To clarify we added the following line: "Exactly the same result is not expected as the distribution of particles for remapping is different and the remapping method tested in section 4.2 does not require the flux correction scheme. However, the similarity in the results indicates that the flux correction scheme works effectively and does not introduce significantly more numerical diffusion."

- *Figure 11 shows lower ice concentration elements at the edge of the ice wake (northeast of the island). Is this expected? Or is this an artifact of the diffusive nature of the remapping?*

– Yes, this is expected. These elements have a lower concentration since these destination elements were only partially covered by sea ice. For clarity we added the following line: "Elements on the edge of the ice pack and the edge of the island wake have concentrations less than one, since during the remapping these elements were not fully covered by source elements."

- *Were thickness distributions examined in any of the two-dimensional cases? For example, in the island case would we expect to see ice thickness growth as ice collides with the island (opposite of the wake side)?*

  – A ridging scheme is in development but has not been validated yet. Simulations here include only elastic repulsion between elements and no thickening of ice occurs in the windward side of the island. For clarity we have added the following text: "Since the contact model used here consists of elastic repulsion without a representation of ridging, ice does not build up on the windward side of the island."

**1.2   Technical Corrections**

- *In the background discussion on pressure ridging, lines 31-42, could also mention two additional Hopkins manuscripts, (1) On the ridging of intact lead ice and (2) On the mesoscale interaction of lead ice and floes.*

  – We added the following text: "Hopkins (19940 developed a two-dimensional cross-sectional model of a single pressure ridge, which was used by Hopkins (1996) to derive a contact model for a two dimensional DEM sea-ice model representing the plastic deformation produced during ridge formation. This model was then used to derive sea-ice yield curves. Hopkins (2004), used the same contact model in sea-ice simulations where the ridging process was represented through a remapping scheme of overlapping converging neighboring elements."

- *Line 133 mentions that "Source element i has its effective area polygon, Pi, advecting with it, ". I assume this includes translation and rotation?*

  – Yes, we added "(including both translation and rotation)" to this line.

- *It would be helpful to add the x and y axis scales to Figure 5, 8, 11, and 12.*

  – We have added x,y scales to figures 5, 8, 11, 12.

**2   Response to reviewer 2 comments**

**2.1   Major comments:**

1. *Lines 32-33 (Capturing the relevant physics of this process using DEM sea ice models has proven to be a challenge): Id strongly recommend citing the works by Marnix van den Berg and colleagues, related to their non-smooth DEM (NDEM) developed at NTNU, in particular, https://doi.org/10.1016/j.coldregions.2018.07.001 and http://dx.doi.org/10.1016/j.marstruc.2019.01.011 (its not self-advertisement, Im not related to the work of NTNU group). One of several advantages of NDEM is that it produces realistic forces during simulations of compliant floe-floe and floe-structure contacts, for floes of any shape  a feature very important from the point of view of problems discussed in the present manuscript (and, especially, for the future developments of DEMSI).*

   – We have added "A related recent development is the non-smooth DEM (NDEM) sea-ice model developed by van den Berg et al. (2018, 2019), which for floes of any shape reproduces realistic forces for compliant floe-floe interactions." to this paragraph.

2. *Related to the previous comment: The contact forces and overlaps computed from the Hookean model are known to have several serious limitations, in particular they are extremely sensitive to the time step used in simulations and to the time stepping method used. Those issues are not the main focus in this manuscript, of course, but they should be mentioned.*

– We added the line "While the Hookean contact model has several limitations, such as sensitivity to the time step and to the time stepping method used, it is sufficient to demonstrate the efficacy of the remapping method.".

3. *As far as I know, even if 2D simulations are performed in LAMMPS, the model assumes that the discrete elements are spheres, not disks. In particular, contact forces are computed for circular, not rectangular contact area, element mass is proportional to r3, not r2, etc. Did the authors implement disk shaped elements in their model and forgot to mention that in the text, or are the simulations performed with the original LAMMPS code? If the second is true, please state it clearly and explain the consequences.*

– We have indeed implemented changes to LAMMPS to allow for treatment of two-dimensional disks rather than spheres moving in a two-dimensional plane. First, as the reviewer correctly points out, the mass of elements by default is proportional to radius$^3$. The new set density/disc command in LAMMPS addresses this, allowing the mass to be proportional to radius$^2$. However, this is not particularly relevant for most DEMSI simulations, since the mass can be set independently and arbitrarily for individual elements. Second, the moment of inertia for two-dimensional simulations in LAMMPS was originally a default of $0.4 \times$mass$\times$radius$^2$, which corresponds to the moment of inertia of a sphere. It is now possible to use a value of $0.5 \times$mass$\times$radius$^2$, which corresponds to the moment of inertia of a cylinder about its axis (i.e. disc), using the new disc option to the LAMMPS verlet integrator (fix nve/sphere). Finally, we have continued to use circular rather than rectangular (or polygonal) elements for computational expediency, since DEMSI aims to enable long time-scale simulations for climate modeling; the issues with this approximation will be discussed in a separate future paper, as they are not strictly related to 2D vs 3D representations or remapping.

4. *Is the angular momentum equation in LAMMPS turned off? Only the linear momentum equation is mentioned in the text.*

– Angular momentum is included in the simulations solving the equations of motion. We expand the description of the momentum equations to include the angular momentum equation.

5. *After having read the whole manuscript and my own comments above, my conclusion is that the significant weaknesses of this work are mostly related to the particular DEM used to perform the computations illustrating the remapping method, rather than to the remapping method itself. Therefore, I have two suggestions:*

(a) *The title of the manuscript is misleading, because it suggests that a new sea ice DEM is presented, which is not the case. In fact, it is the very opposite: the DEM used here is much more primitive than several published ones, including those cited in the introduction, and it is not suitable to simulate ridging processes due to the limitations of the very simple contact model used (mentioned above) and several other limitations of LAMMPS (not mentioned and, unfortunately if the authors are planning to use that model in the future hard or even impossible to overcome). Id suggest changing the title to something like Geometric remapping of particle distributions for discrete element sea ice models, so that it is clear what the paper contains and what its strengths are. Note that the acronym DEMSI is not used even once in the main text! (only in the abstract and acknowledgements). I know that GMD requires a name for the model being presented, but here its the remapping algorithm that deserves a name, not the sea ice DEM!*

– We disagree with the reviewer on this point. We believe the title, "Geometric remapping of particle distributions in the Discrete Element Model for Sea Ice (DEMSI v0.0)", does not imply that a whole new DEM model is presented, only that a remapping method is presented. This geometric remapping method is implemented as an integral part of the DEMSI code base, as available from the "code availability" section in the paper. We believe that, for the applications DEMSI is intended for, some form of remapping techniques will be an integral part of running a sea-ice DEM model. We are developing more physically appropriate contact

models for sea ice, which will be presented in the future, but believe the DEM contact model is only one aspect of the development work needed to make a successful sea-ice DEM model for the application DEMSI is designed for. We add the line "This remapping method forms part of a new sea-ice dynamical core, the Discrete Element Model for Sea Ice (DEMSI), currently under development for use in coupled Earth system models." to the last paragraph of the introduction.

(b) *State it clearly in the last part of the introduction that the performance of the new method is illustrated based on simple simulations with a DEM that can be replaced with a serious one later on, and that the limitations of the DEM are not very crucial for the cases analyzed (in fact, the first 1D and 2D tests with translation and/or rotation of elements do not require any momentum equations/contact models, do they?).*

– We add the following text to the final paragraph of the introduction: "This remapping method forms part of a new sea-ice dynamical core, the Discrete Element Model for Sea Ice (DEMSI), currently under development for use in coupled Earth system models. The test cases used here to illustrate the method use a greatly simplified DEM contact model which is sufficient to demonstrate the efficacy of the remapping method. Work is ongoing developing a contact model for DEMSI representing sea-ice dynamics and including a representation of sea-ice fracture and ridging."

**2.2 Minor, mostly technical comments:**

1. *Lines 44-48: Id suggest changing collisions to contacts everywhere in this fragment, as not all contacts are of the collision-type (especially during ridging, of interest here). (The term collision detection actually comes from early molecular-dynamics type models, where the discrete elements were assumed infinitely rigid, so that each contact was by definition a short-lived collision.)*

– We have replaced "collision" with "contact" throughout.

2. *Lines 315-316: there are no empty elements, representing open water. Actually, why not? What happens when the ice from a given polygon disappears, is that element removed from the simulation? Would it be advantageous to have no-ice polygons? It is discussed briefly around line 380, but it is assumed that those effects are relevant at the ice edge only. What about situations when many open water patches are present within the ice cover? Would the conclusion regarding unimportance of open water elements remain the same?*

– Once an element contains no ice it is removed from the simulation. A major reason not to include empty elements is computational. If empty elements were included, the whole domain would have to be covered in elements and the number of elements in the simulation would be significantly larger, and the computational speed of the simulation significantly slower. We do not believe it would be advantageous to have empty elements - it is unclear what equation of motion these elements would obey (stationary?) or how they would interact with other elements. Given the minimal effect the addition of empty pseudo-elements has on the edge concentration we do not believe they would affect the calculation significantly in patches within the ice. We note that ice filled elements also have a fraction of open water associated with them. We modify this section to read: "For DEM sea-ice models, elements only exist where sea ice exists, with elements that loose all their ice being removed from the simulation. One advantage of this is that fewer elements are needed than Eulerian cells (which must cover the whole domain), increasing computational performance for the DEM model. Consequently, there are no empty elements representing open water to be included in tracer gradient calculations, or in the neighboring element tracer extrema used to limit those gradients."

3. *(47) and line 348: The symbol c is used here for the drag coefficient, and has been used earlier as ice concentration. cw is used for drag coefficient further in line 356.*

– We have added appropriate subscripts to the drag coefficients for clarity.

4. *Line 414: 500 m/s?*

   – We rewrite the descriptions for the 1D and 2D remapping test cases in terms of the element translation distance per remapping, rather than element velocity, since this is the physically important quantity.

5. *Figure 5 and the following colour figures: explain in their captions what the colour scale shows.*

   – We added "Ice concentration" to the color bars in Figures 5, 8, 11, and 12, and to the y axis of Figures 3, 6, and 9. We modified the sub-figure titles for Figure 4 clarifying what is plotted. We add "(fraction of elements covered in sea ice)" as a definition to ice concentration in the caption of Figure 5, and add "sea-ice concentration" to the caption of Figure 8.

[revised manuscript text omitted]